# Impact of the suboptimal communication network environment on telerobotic surgery performance and surgeon fatigue

Harue Akasaka[1], Kenichi Hakamada[1,2]*, Hajime Morohashi[1,2], Takahiro Kanno[2,3‡], Kenji Kawashima[2,3,4‡], Yuma Ebihara[2,5‡], Eiji Oki[2,6‡], Satoshi Hirano[2,5‡], Masaki Mori[2,7‡]

1 Department of Gastroenterological Surgery, Hirosaki University Graduate School of Medicine, Hirosaki, Japan, 2 Committee for Promotion of Remote Surgery Implementation, Japan Surgical Society, Tokyo, Japan, 3 RIVERFIELD Inc., Tokyo, Japan, 4 Department of Information Physics and Computing School of Information Science and Technology, The University of Tokyo, Tokyo, Japan, 5 Department of Gastroenterological Surgery II, Hokkaido University Faculty of Medicine, Sapporo, Japan, 6 Department of Surgery and Science, Kyushu University, Fukuoka, Japan, 7 Tokai University School of Medicine, Isehara, Japan

☯ These authors contributed equally to this work.
‡ These authors also contributed equally to this work.
* hakamada@hirosaki-u.ac.jp

**Data Availability Statement:** All relevant data are within the paper and its Supporting Information files.

## Abstract

### Background

Remote surgery social implementation necessitates achieving low latency and highly reliable video/operation signal transmission over economical commercial networks. However, with commercial lines, communication bandwidth often fluctuates with network congestion and interference from narrowband lines acting as bottlenecks. Therefore, verifying the effects on surgical performance and surgeon fatigue when communication lines dip below required bandwidths are important.

### Objectives

To clarify the communication bandwidth environment effects on image transmission and operability when bandwidth is lower than surgical robot requirements, and to determine surgeon fatigue levels in suboptimal environments.

### Methods

Employing a newly developed surgical robot, a commercial IP-VPN line connected two hospitals 150 km apart. Thirteen surgical residents remotely performed a defined suturing procedure at 1-Gbps to 3-Mbps bandwidths. Communication delay, packet loss, time-to-task completion, forceps-movement distance, video degradation, and robot operability were evaluated before and after bandwidth changes. The Piper Fatigue Score-12 (PFS-12) was used to measure fatigue associated with surgeon performance.

**Funding:** This work was supported by a grant from the Japan Agency for Medical Research and Development (AMED, https://www.amed.go.jp/), Grant Number JP21hs0122001h0002.The funders had no role in study design, data collection and analysis, decision to publish, or preparation of the manuscript.

**Competing interests:** I have read the journal's policy and the authors of this manuscript have the following competing interests: Kenji Kawashima is employed by RIVERFIELD and owns stock in the company. Takahiro Kanno is employed by RIVERFIELD and serves as CTO. Harue Akasaka and other co-authors have no conflict of interest. This does not alter our adherence to PLOS ONE policies on sharing data and materials.

## Results

Roundtrip communication time for both 1-Gbps and 3-Mbps lines averaged 4 ms. Video transmission delay from camera to monitor was comparable, at 92 ms. Surgical robot signal transmission rate averaged 5.2 Mbps, so changing to 1-Gbps-3-Mbps lines resulted in significant packet loss. Surgeons perceived significant roughness, image distortion, diplopia, and degradation of 3D images ($p = 0.009$), but not changes in delay time or maneuverability. All surgeons could complete tasks, but objective measurement of task-completion time and forceps-travel distance were significantly prolonged ($p = 0.013$, $p = 0,041$). Additionally, PFS-12 showed post-procedure fatigue increase at both 1-Gbps and 3-Mbps. Fatigue increase was significant at 3-Mbps ($p = 0.041$).

## Conclusions

In remote surgery environments with less than the optimal bandwidth, even when delay time and operability are equivalent, reduced surgical performance occurs from video degradation from packet loss. This may cause increased surgeon fatigue.

## Introduction

Telesurgery holds great promise as a mean way to improve accessibility to high quality healthcare in areas with limited healthcare resources and as a new method in surgical education [1]. In 2001, the world's first inter-Atlantic telesurgery was successfully performed [2] and a telesurgery system was established in Canada [3]. However, the social implementation of telesurgery was deferred for a long time due to the following: the communication delay time was too long for clinical application with communication lines at the time [4,5], secure dedicated lines were too expensive for most clinical use [6], communication security could not be ensured over internet lines [7], so for various reasons, there was minimal development of new robots for telesurgery [8].

These days, issues of communication delay and security, which were key barriers to remote surgery, are being resolved by the development of high-speed, large-capacity communication networks and advances in information processing technology. In addition, the patent for the surgical robot system of the preceding generation has expired, which has led to the development of new types of surgical robots and the resumption of social demonstration studies of remote surgery [8–11]. Our group conducted a telesurgery experiment in which two hospitals located 150 km apart were connected by a commercial line. A surgical robot under development was used, and a telesurgical operation was possible without significant delay [1].

Remote surgery is performed by transmitting both video signals and control signals for robot operation. Of these, the number of signals which are required to operate the robot arm and control energy devices is small and all can be transmitted over narrow-band communication lines. Video signals, on the other hand, contain a large amount of information and require a bandwidth of more than 1.5 Gbps for full high vision images and more than 6 Gbps for 4K images. This presents the difficulty of achieving uncompressed transmission with the communication bandwidth of ordinary commercial lines. For this reason, compressed transmission of video signals is essential. However, since the relationship of the compression ratio and the delay time is the trade-off, the degree of compression needs to be limited. Also, excessive compression causes video degradation. Therefore, preparing a line with a sufficient communication bandwidth that can transmit the video signal satisfactorily, after compression, is necessary.

Among the currently available commercial lines, a line that guarantees a wide bandwidth is expensive and is not suitable for general clinical use from the viewpoint of economic efficiency [6]. The best effort type line, however, for which the maximum communication capacity is openly stated, is highly economical. Some unfortunate realities of using the best effort type line, are that bandwidth may temporarily become smaller than the required communication line bandwidth when the line is congested, or that bandwidth narrower than the required bandwidth, may bottleneck when multiple lines are connected. In such network environments, with less-than-optimal communication conditions, packet loss and communication delay of transmission signals can occur. In the case of packet loss, video transmission containing a large amount of communication information is greatly affected. The degree of packet loss varies greatly from the degree at which a surgeon cannot detect any change in the image to the degree that the image is clearly changed [12].

Surgical robot systems have a minimum required bandwidth for transmission of video signals and operation signals. However, the actual degree of change to the video is unclear, as is the effect on surgical performance when communication bandwidth dips below the system's minimum bandwidth. In addition, physiological effects such as surgeon fatigue have not been investigated thus far. In this study, we examined the effects on surgical performance and surgeon fatigue of a communication environment with a bandwidth narrower than the required bandwidth during an actual remote surgery experiment using a commercial line.

## Methods

### Communication environment

Hirosaki University Hospital and Mutsu General Hospital, located about 150 km apart, were connected through a commercial bandwidth-guaranteed line provided by NTT East (NIPPON TELEGRAPH AND TELEPHONE EAST CORPORATION, TOKYO, JAPAN) (Fig 1). The communication bandwidth was initially set at 1-Gbps, and then changed to 3-Mbps. We evaluated the latency of each communication line, the latency associated with signal compression/decompression processing, and the overall latency of video transmission between the laparoscopic camera and the monitor.

For CODEC, we used Encoder Zao-SH and decoder Zao-View of Soliton Systems (Tokyo, Japan). This CODEC has a function that reserve 1Mbps of communication bandwidth for robot operation signals preferentially and allocates the remaining bandwidth to video transmission.

### Surgical robots and tasks

Using a surgical robot being developed by RIVERFIELD, Inc. [13], thirteen surgical residents, ranging from 2nd to 4th year, who had no experience in robotic surgery, but had robotic procedure experience in actual robotic surgery as assistants, performed suture ligation of an intestinal model by remotely manipulating robotic arms with a controller while viewing the surgical field image on an open 3D monitor. To avoid the influence of habituation, the subjects were divided into two groups: seven subjects performed the procedure in the sequence of 1-Gbps, then 3-Mbps, and the remaining six subjects performed the procedure in the order of 3-Mbps, then 1-Gbps. For both groups, a 10-minute practice period was provided immediately before task performance measurement (Fig 2).

The intestinal model was marked to indicate suture sites at 5 mm intervals, and the operation was performed with five sutures and three ligatures in each suture. Operation performance was evaluated by measuring the task completion time from the start of intestinal

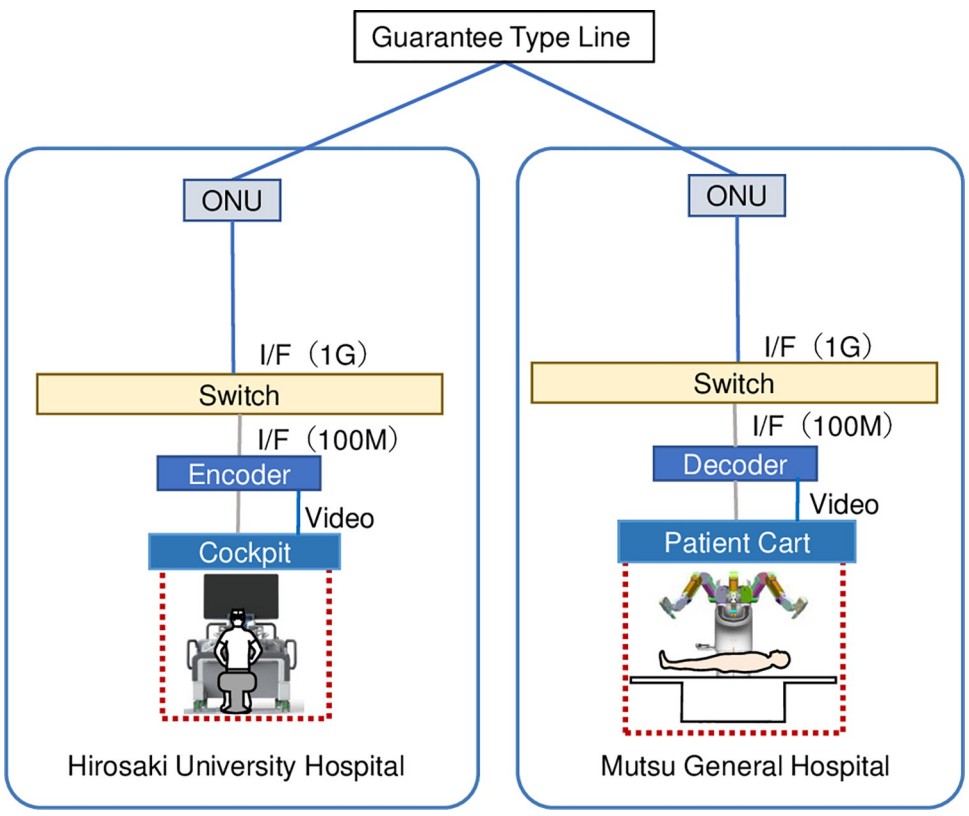

**Fig 1. Network system.** Hirosaki University Hospital and Mutsu General Hospital, located about 150 km apart, were connected through a commercial bandwidth-guaranteed line. OUN: Optic network unit, CPE RT: Customer premises equipment remote terminal, I/F: Interface.

anastomosis to the end of three ligatures of the final suture, and the distance traveled by the robotic arm forceps during the operation.

Furthermore, using the Global Evaluative Assessment of Robotic Skills (GEARS) score devised by Goh et al. [14] (S1 Table), two robotic expert surgeons evaluated the subject performance (depth of perception, bimanual dexterity, efficiency, force sensitivity, autonomy, robotic control). The mean values of the two evaluators were compared for each line band.

### Subjective evaluation of the examinee

**Modified System Usability Scale.** The m-SUS was created by modifying the System Usability Scale (SUS) [15], which was developed to subjectively evaluate the usability of newly constructed systems, and the usability of the telesurgery system was evaluated (Fig 3). Nine items were rated on a 5-point scale in descending order of usefulness, and the total score was shown as the score.

**Robot Usability Score.** In order to evaluate the usability of the operation of the robot system, we created the Robot Usability Score, which evaluates eight items: physical comfort, manual operability, foot pedal operability, stereoscopic performance, forceps operability, smoothness, satisfaction, and effectiveness (Fig 4). The usability was evaluated on a 5-point scale in descending order of usefulness, and the average values for each item were compared by line.

**Image quality evaluation scale.** In order to evaluate whether image quality degradation due to changes in line conditions interferes with the performance of the procedure, an image

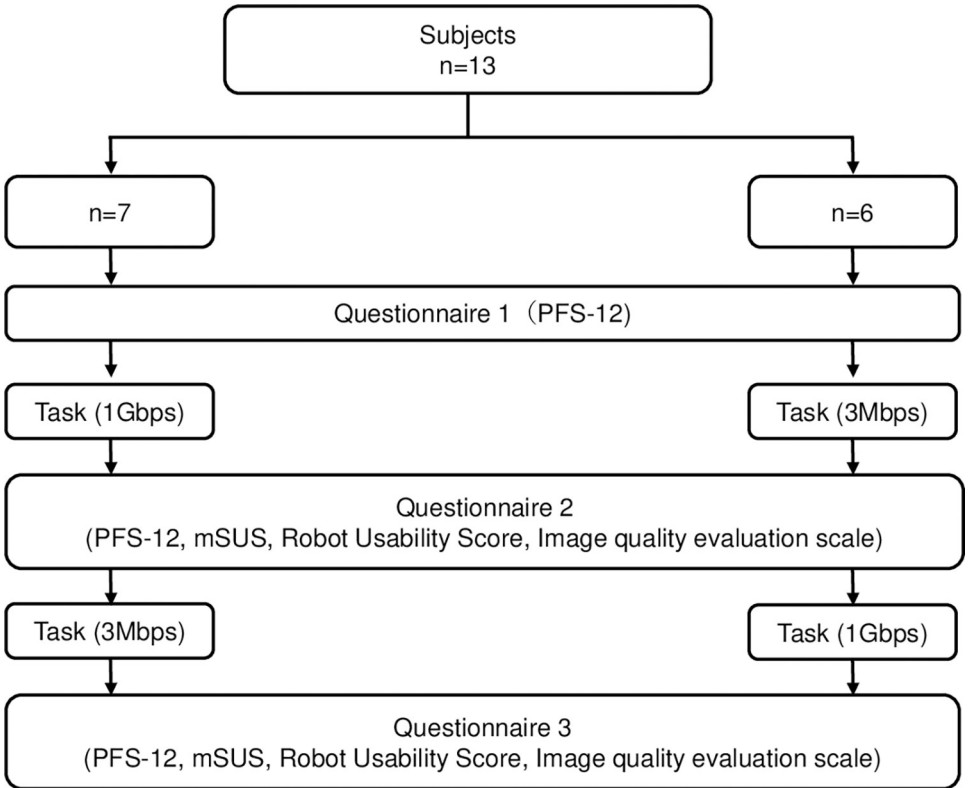

**Fig 2. Flowchart of the order of the line speed and subjective evaluation of the task.** The subjects were divided into two groups: Seven subjects performed the procedure in the sequence of 1-Gbps, then 3-Mbps, and the remaining six subjects performed the procedure in the order of 3-Mbps, then 1-Gbps. They answered questionaries pre- and post-procedure for each bandwidth.

quality evaluation scale was developed (Fig 5). A five-point scale was used, with a high score indicating no deterioration in image quality and no effect on the performance of the procedure. The median score was compared by line type.

**Piper Fatigue Scale-12 (PFS-12).** The subjective fatigue of the surgeons was assessed using the Piper Fatigue Scale-12 (PFS-12) survey [16] (S2 Table). the PFS-12 consists of 12 questions on fatigue and is divided into four subscales: behavioral, emotional, sensory, and cognitive. The scores are rated on a 10-point scale in descending order of fatigue, and the method of score calculation is reported by Reeve et al. The questionnaire was filled out immediately before and after the procedure, and the changes before and after the procedure were measured. Pre- and post-procedure changes were measured.

**Statistical analysis.** Data are presented as frequency for categorical data and as mean ± standard deviation or median (range) for continuous data. EZR software was used for analysis [17]. Normality was tested with the Shapiro-Wilk test, and if normality was not rejected, the paired Student's t-test was used. When normality was rejected, the Wilcoxon test was used. Statistical significance was determined at $p < 0.05$.

## Ethics statements

There is no need to address this or provide an ethics application as this research does not fall under the category of "life science/medical research on human subjects." The surgeons were using robotic technology to suture inanimate objects in a simulated telesurgery situation. With

## Modified System Usability Scale

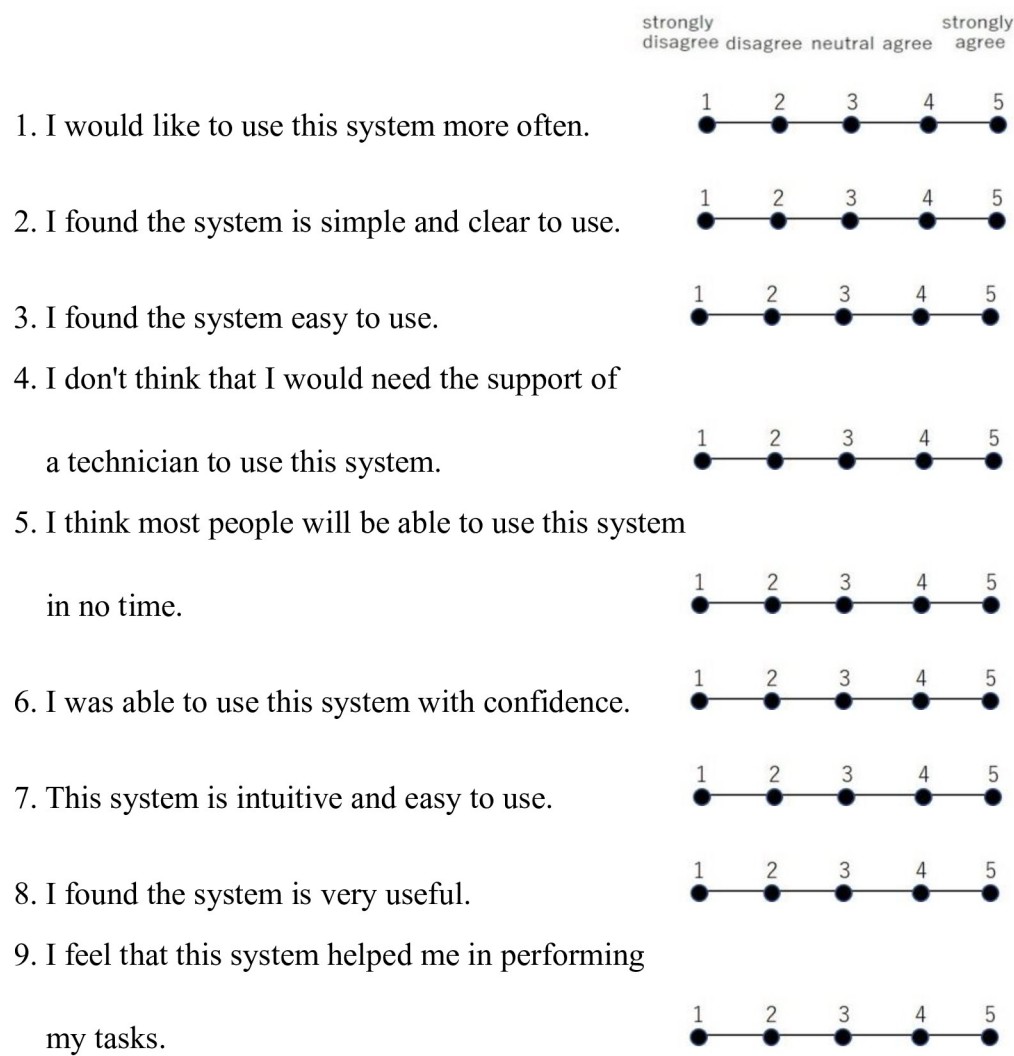

**Fig 3. Modified System Usability Scale.**

regard to the above, we obtained a formal waiver from Hirosaki University Ethics Committee for this study. All subjects (participating surgeons) were informed about the study in writing, and we obtained their written consent.

## Results

### Transmission delay time

Communication delay was comparable with a median of 4 ms (range 3–7) for the 1-Gbps line and 4 ms (3–8) for the 3-Mbps line. Adding 61 ms (48–74) for encoder and decoder delay to the communication delay, the delay caused by the remote surgical operation was 65 ms (51–81) for the 1-Gbps line and 65 ms (51–88) for the 3-Mbps line. The glass-to-glass time, which also includes the time required for the laparoscopic camera to process images of the surgical

## Robot Usability Score

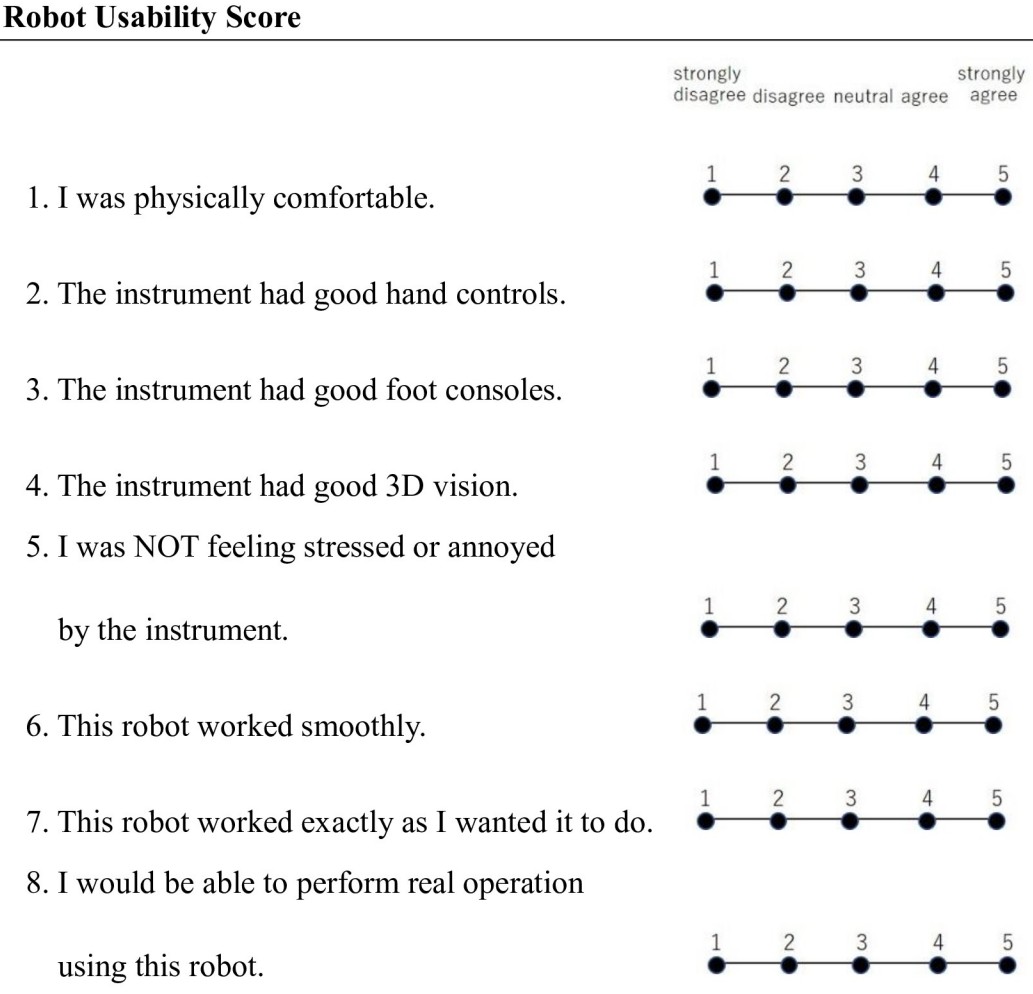

**Fig 4. Robot Usability Score.**

field and the reaction time of the surgical monitor, was 92 ms (73–117) for the 1-Gbps line and 92 ms (73–118) for the 3-Mbps line.

### Video signal packet loss and changes in surgical field images

We simultaneously and continuously measured the communication bandwidth of the transmission signal, frame rate of the video, packet loss, and communication delay time during the surgical task. A typical measurement example is shown in Fig 6. When a line with a communication bandwidth of 1-Gbps was used, the video signal transmission rate was stable at 5.2 Mbps (range 4.8–5.3), and the average packet loss rate was as low as 0.042%. However, when the communication bandwidth was changed to 3-Mbps, the average packet loss rate increased to 4.82%. When a subject started a robotic operation, the video communication bandwidth decreased to 2 Mbps because CODEC allocated 1 Mbps bandwidth to the robot operation signal on a priority basis (arrowhead in the Fig 6). On the contrary, when the surgical operation ended, the degree of packet loss was reduced (arrow in the Fig 6). The communication delay (round-trip transmission time: RTT) was stable at less than 3 ms regardless of the bandwidth. The same results were obtained in all 13 subjects.

### Image quality evaluation scale

Please rate the quality of the robotic surgery images on a scale of 1-5

according to the following criteria.



[Evaluation criteria]

5: No degradation in image quality

4: Image quality degradation is there, but it doesn't bother me.

3: Image quality is degraded but does not interfere with the procedure

2: Image quality degradation slightly hinders the procedure.

1: Image quality degradation makes it impossible to perform the procedure

**Fig 5. Image quality evaluation scale.**

### Evaluation of surgical performance

Task completion time was significantly prolonged with the slower 3-Mbps line (350.2 ± 91.7s vs. 386.2 ± 92.6s, p = 0.013). Forceps travel distance was significantly longer for right-handed forceps manipulation (3,786.7 ± 750.8 mm vs. 4,064.2 ± 952.2, p = 0.041), but left-handed forceps manipulation also showed a trend toward longer distance (3,743.0 mm ± 480.6 vs. 4,028.2 ± 742.8, p = 0.10). Evaluation of surgical performance by GEARS score, however, showed no significant difference between the bands (25.4 ± 3.4 vs. 23.6 ± 2.6, p = 0.12). Among the evaluation items, the accuracy of forceps movements and sutures were evaluated under the category of "Depth perception," and it was shown as being equal by bandwidth (3.7 vs. 3.5, p = 0.35).

### Usability of the surgical robot

The evaluation of the usefulness of the tele-robotic surgery system by m-SUS showed a lower trend at 3-Mbps than at 1-Gbps (21.31 ± 4.50 vs. 19.08 ± 4.55, p = 0.062).

In the Robot Usability Score, the evaluation of the stereoscopic view was significantly lower for the 3-Mbps line than in the 1-Gbps line (p = 0.009), and hand control operation tended to be lower (p = 0.06). There was no difference by line bandwidth in overall comfort of the surgical robot, smoothness of robot operation, follow-up, effectiveness (Table 1).

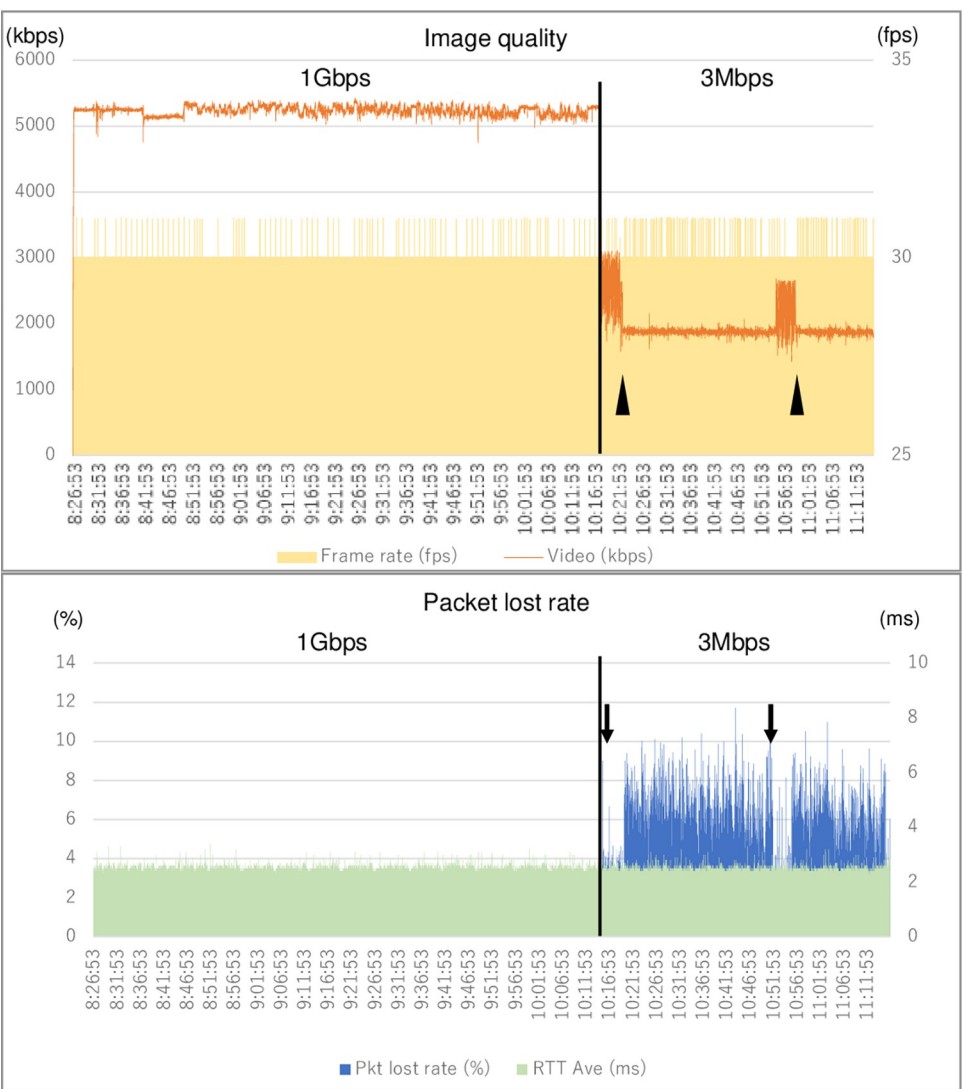

**Fig 6. An example of the bandwidth, frame rate of the video, packet lost, and delay time.** The video signal transmission rate was steady at 5.2 Mbps, and the average packet lost rate was as low as 0.042% with 1 Gbps. The average packet loss rate increased to 4.82% with 3 Mbps. When a subject began a robotic operation, the visual communication bandwidth was reduced to 2 Mbps because CODEC assigned a priority of 1 Mbps to the robot operation signal (arrowhead). The surgical procedure was completed, the amount of packet loss decreased (arrow). Regardless of bandwidth, the communication delay was less than 3 ms, Framerate (fps): Framerates per second, Video (kbps): Video transmission value loss (kilobits per second), Pkt lost rate: Packet lost rate.

## Subjective image quality evaluation

Video abnormalities such as screen coarseness (4/13), diplopia (3/13), horizontal line delineation (1/13), and stereoscopic difficulty (2/13) were observed on the 3-Mbps line. In terms of the effect of image quality degradation on the performance of the procedure, evaluation by the image quality rating scale showed that the 3-Mbps line was significantly lower than the 1-Gbps line (4 (4–5) vs. 3 (2–4), p = 0.034).

## Self-reported fatigue score

On the PFS-12 score, the overall score increased significantly after the procedure for both 1-Gbps and 3-Mbps lines (Table 2A). The 3-Mbps line, in particular, showed an increase on all

**Table 1. Comparison of Robot Usability Score across line bandwidths.**

| Speed of the line | 1-Gbps | 3-Mbps | P-value[a] |
|---|---|---|---|
| Physical Comfort | 3.07 ± 0.95 | 2.77 ± 1.01 | 0.28 |
| Hand Control | 3.15±0.90 | 2.46 ± 0.97 | 0.06 |
| Foot Control | 4.08 ± 0.64 | 4.07 ± 0.75 | 1.00 |
| 3D Vision | 3.31 ± 0.85 | 2.38 ± 1.04 | 0.009[b] |
| Annoyed or stressed | 2.76 ± 0.83 | 2.62 ± 0.87 | 0.62 |
| Smoothness | 2.85 ± 0.90 | 2.85 ± 0.90 | 0.93 |
| Satisfaction | 2.84 ± 0.99 | 2.69 ± 0.85 | 0.88 |
| Reality | 3.00 ± 1.00 | 2.61 ± 1.04 | 0.28 |

The evaluation of the stereoscopic view was significantly lower for the 3-Mbps line than in the 1-Gbps line. There was no statistical difference of the score in the other parameters.

[a] p values were calculated using the paired *t*-test.

[b] $p < 0.05$.

four subscales. On the contrary, the difference in the overall score of fatigue before and after the procedure on the 3-Mbps line (Δ3 Mbps) was significantly larger than that of the 1-Gbps line (Δ1 Gbps) (Table 2B).

## Discussion

In this study, slight disturbances to the video image degraded the surgical performance and increased surgeon fatigue, even though the transmission delay was minor and operability did not change in a communication environment with a bandwidth lower than that required for video transmission from a surgical robot. Since the required bandwidth of surgical robots varies from model to model depending on the number of video pixels and data compression

**Table 2. Comparison of the Piper Fatigue Scale-12 (PFS-12) between line bandwidths.**

(a) PFS-12 score values before and after the task for each line.

| Speed of the line | 1 Gbps | | | 3 Mbps | | |
|---|---|---|---|---|---|---|
| | before | after | P-value [a] | before | after | P-value [a] |
| All scores | 2.93 | 3.60 | 0.049 [b] | 2.16 | 3.54 | 0.002 [b] |
| Behavioral | 2.51 | 3.53 | 0.024 [b] | 1.17 | 3.3 | 0.004 [b] |
| Affective | 3.23 | 3.87 | 0.070 | 2.66 | 4.12 | 0.016 [b] |
| Sensorial | 3.26 | 3.92 | 0.180 | 2.79 | 3.72 | 0.015 [b] |
| Cognitive | 2.69 | 3.10 | 0.388 | 1.46 | 2.99 | 0.016 [b] |

(b) Comparison of the change in PFS-12 values before and after the procedure for each line.

| Speed of the line | ⊿1 Gbps | ⊿3 Mbps | P-value [a] |
|---|---|---|---|
| All scores | 0.68 ± 1.12 | 1.37 ± 1.23 | 0.041 [b] |
| Behavioral | 1.03 ± 1.43 | 1.59 ± 1.64 | 0.062 |
| Affective | 0.64 ± 1.17 | 1.46 ± 1.89 | 0.126 |
| Sensorial | 0.64 ± 1.64 | 0.92 ± 1.18 | 0.471 |
| Cognitive | 0.41 ± 1.65 | 1.53 ± 1.37 | 0.084 |

The all score increased significantly after the procedure for both 1-Gbps and 3-Mbps lines. The 3-Mbps line, in particular, showed an increase on all four subscales.

The difference in the all score of fatigue pre- and post-procedure on the 3-Mbps line (Δ3 Mbps) was significantly larger than that of the 1-Gbps line (Δ1 Gbps).

[a] p values were calculated using the paired *t*-test.

[b] $p < 0.05$.

ratio, the results of this study indicate the necessity of measuring the required bandwidth of the surgical robot to be used in advance and preparing a communication line with at least the minimum communication bandwidth.

In general, a dedicated line that guarantees the use of a wide bandwidth maintains the maximum communication bandwidth and has excellent stability and speed for high-capacity communication. However, the cost is remarkably high, making unsuitable for the widespread social implementation of remote surgery [6]. Conversely, the best effort type line, which advertises the maximum communication bandwidth among closed circuits with guaranteed security, is inexpensive and widely used, but has the risk of falling below the minimum bandwidth because the available bandwidth is affected by communication congestion. For this reason, new, more economical line services have been launched, such as lines with secured minimum bandwidth (burst lines) and best effort lines with bandwidth so wide that it surpasses instability.

Nevertheless, the available communication infrastructure varies from country to country and region to region. When promoting the social implementation of telesurgery, several different lines may be connected, and the possibility cannot be excluded that the amount of communication bandwidth may change due to the existence of bottlenecking, which creates narrow bandwidth in some parts of the network, and that the bandwidth may fall below the required level. Therefore, it is necessary to consider in advance the impact on surgical performance that changes in the communication environment will have when conditions fall below the required bandwidth. In addition, since surgeons adapt under stress even when they encounter a less than optimal environment [18,19], including the degree of fatigue surgeons face is a necessary part of constructing an evaluation.

In this study, the surgical robot required a video communication bandwidth of 5.2 Mbps, with a 1-Gbps bandwidth-guaranteed line, the communication delay was 4 ms and the video transmission delay from the endoscope camera to the monitor projection was 92 ms, with no problems in 3D video transmission and good operability. On the other hand, with the 3-Mbps line, the communication delay and video transmission delay were the same without extension, but the average packet loss was 4.82%. As a result, the video was degraded and the Robot Usability Score evaluated the 3D image quality as significantly low. Although they were able to complete all surgeries, the surgeon had to perform accurate suture ligation maneuvers in a suboptimal video environment, resulting in longer forceps travel distances and longer task completion times. The significantly longer forceps travel distance in the right hand than in the left hand was presumably due to the fact that the needle driver was attached to the robotic arm of the right hand when performing the suturing task.

In remote surgery, it is known that delays occur in each step of communication network transmission, information compression and decompression processing, video signal conversion at the camera, and monitor projection, resulting in prolonged task completion time and increased task error rate [19,20]. Furthermore, communication fluctuations (jitter) and packet loss are known to cause video degradation [12] and make equipment operation difficult [21]. The CODEC used in this study has a mechanism that preferentially allocates 1 Mbps for operation signal transmission, yet no obstacle to the operability of the robot on either the m-SUS or Robot Usability Score evaluations. In addition, the video transmission delay is extremely short, 92 ms, which is less than 200 ms, which is considered to be an acceptable delay time for teleoperations [18,22]. In addition, since the delay was less than 100 ms, which is considered to be an acceptable delay time for telesurgery [22,23], the main cause of the prolongation of the task completion time in this study was assumed to be caused by the packet loss of the video transmission, not the delay. In a previous study, a telesurgery experiment was conducted using a 5 Mbps line with the same robot and the same facility conditions, and packet loss of less than 1% with no video degradation was observed [24]. Based on this information, we chose to conduct

this experiment using a 3-Mbps line. Video degradation, which can be caused by packet loss, varies, so there is a need to verify the acceptable level of packet loss for each surgical robot.

The Peak signal-to-noise ratio (Peak) method [25,26] and the structural similarity index measure (SSIM) [27] are known as objective evaluation methods for video degradation. However, it is also pointed out that they do not necessarily reflect subjective evaluation [28]. Moreover, the most important aspect of the video quality required for remote surgical images is whether it is good enough for the surgeon to be able to perform the operation. Therefore, we evaluated the image quality in terms of the outcome as to whether it interfered with the procedure or not. As a result, the 3-Mbps line was evaluated as not interfering with the procedure, although video degradation was perceived in comparison with the 1-Gbps line, yet the video degradation was within the range to which the surgeon could adapt. In an objective assessment, completion time was prolonged, which indicated that video degradation certainly had an impact on surgical performance, but it was not recognized to be a serious obstacle to the tasks. In other words, the surgeon adapts to the degradation of the image and subconsciously operates at a slightly slower pace to maintain accuracy and complete the task, so when asked, in the surgeon's conscious personal experience, the operation is unaffected. Objectively, however, the task completion time is longer. These facts also suggest that there could be a discrepancy between objective and subjective assessments.

On the other hand, it stands to reason that a certain level of burden might be imposed upon the surgeon when performing operations in such a suboptimal environment. Therefore, we also evaluated the degree of surgeon fatigue due to changes in the communication environment. In the past, it was reported that surgeons were able to adapt to an extended transmission delay by training for this sort of delay time [18]. In other words, surgeons were able to adapt to less-than-optimal conditions by expending a certain amount of energy. It is noteworthy that surgeon fatigue would increase in the early stages of such changing conditions. According to the PFS-12 survey, surgeon' fatigue increased before and after the operation on both lines, but the increase was particularly significant with the 3-Mbps line. This was thought to be a result of the energy spent dealing with the video degradation caused by the change of bandwidth.

Armijo et al. previously reported the difference in fatigue between laparoscopic and robotic surgery by measuring upper body muscle activity and self-reported fatigue using PFS-12. [29] Also, it was reported that more than half of the surgeons performing robotic surgery for a long period of time experienced physical issues [30]. In this study, we showed that not only the robotic surgery itself but also the video degradation that occurs specifically in teleoperations affected surgical performance and surgeon fatigue. In the future, communication environments will rapidly advance; the telesurgery environment will change to be more stabilize with fewer communication interruptions. However, there is no doubt that constructing a minimum environment necessary for safe and economic telesurgery is needed. Furthermore, the telecommunications environment is not always sufficient in some of the very regions of the world where telesurgery is badly needed. In order to succeed in such environments, both objective and subjective evaluations are likely necessary to set appropriate standards for acceptable packet loss and video quality, because there will be some discrepancy between subjective and objective evaluations.

This study is meaningful because it was conducted in an actual telesurgery environment using a real commercial line. On the other hand, it also has the following limitations: (1) Communication bandwidth was tested with only two types, a 1-Gbps line and a 3-Mbps line, and the bandwidth changed discontinuously. Therefore, we were unable to evaluate the effect of the bandwidth on the video degradation continuously and quantitatively. However, in a previous study conducted under similar conditions, in which a 5-Mbps line was used, the video image and the task completion time did not change, despite the packet loss that existed.

Hence, the 3-Mbps line is assumed to be the approximate critical bandwidth. (2) Since the required bandwidth varies depending on the types of surgical robots and encoders, the limit of the bandwidth in this study applies only to this environment. (3) The surgical tasks were only basic movements. Since negative effects were observed even for simple procedures in a suboptimal environment, considerable negative effects might be seen in actual surgery. (4) The test subjects were limited to residents who had no experience in robotic surgery. Acceptable limits for video degradation and delay would expand for those with extensive robotic experience. (5) The evaluating methods for surgical performance were subjective. However, using three different metrics: m-SUS, Robot Usability Score, and GEARS, and evaluating the performance quantitatively is surely valuable. In addition, we examined the effects of video degradation on the technique and the fatigue level of the surgeons. They completed the operations despite the energy it consumed, even in a suboptimal operating environment thanks to adaptation. Therefore, evaluating the physiological effects on the surgeon has proven to be important.

## Conclusions

In a communication environment with less than the required bandwidth, video image degrades due to packet loss, the task completion time is prolonged, and this can be a cause of increased surgeon fatigue, even when no prolongation of transmission delay or degradation of operability is seen. Establishing an acceptable standard environment is necessary for performing telesurgery when temporary suboptimal communication conditions are encountered. In addition, the selection of an appropriate communication line is essential for the social implementation of this modality.

## Supporting information

**S1 Table. Global Evaluative Assessment of Robotic Skills (GEARS).**
(DOCX)

**S2 Table. Piper Fatigue Scale-12 (PFS-12).**
(DOCX)

## Acknowledgments

We would also like to express our deepest gratitude to RIVERFIELD Inc, NTT East, Soliton Systems K.K., Hirosaki University, and Mutsu General Hospital for their cooperation in the experiment. We sincerely thank Shari Joy Berman for professionally editing the English draft of this manuscript.

## Author Contributions

**Conceptualization:** Kenichi Hakamada, Yuma Ebihara, Eiji Oki, Satoshi Hirano, Masaki Mori.

**Data curation:** Harue Akasaka.

**Formal analysis:** Harue Akasaka.

**Funding acquisition:** Kenichi Hakamada, Hajime Morohashi, Yuma Ebihara, Eiji Oki, Satoshi Hirano, Masaki Mori.

**Investigation:** Harue Akasaka, Hajime Morohashi, Takahiro Kanno, Yuma Ebihara.

**Methodology:** Harue Akasaka, Kenichi Hakamada, Hajime Morohashi.

**Project administration:** Kenichi Hakamada, Hajime Morohashi, Masaki Mori.

**Resources:** Takahiro Kanno, Kenji Kawashima.

**Software:** Takahiro Kanno, Kenji Kawashima.

**Supervision:** Kenichi Hakamada, Yuma Ebihara, Eiji Oki, Satoshi Hirano, Masaki Mori.

**Visualization:** Harue Akasaka, Kenichi Hakamada.

**Writing – original draft:** Harue Akasaka, Kenichi Hakamada.

**Writing – review & editing:** Kenichi Hakamada, Hajime Morohashi, Takahiro Kanno, Kenji Kawashima, Yuma Ebihara, Eiji Oki, Satoshi Hirano, Masaki Mori.

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
