## [Decision Letter · Decision Letter 0]

1 Mar 2022

PONE-D-22-03175Impact of the suboptimal communication network environment on telerobotic surgery performance and surgeon fatiguePLOS ONE

Dear Dr. Hakamada,

Thank you for submitting your manuscript to PLOS ONE. After careful consideration, we feel that it has merit but does not fully meet PLOS ONE’s publication criteria as it currently stands. Therefore, we invite you to submit a revised version of the manuscript that addresses the points raised during the review process.

We look forward to receiving your revised manuscript.

Kind regards,

Norikatsu Miyoshi, M.D., Ph.D., FACS

Academic Editor

PLOS ONE

Journal Requirements:

2. You indicated that ethical approval was not necessary for your study. We understand that the framework for ethical oversight requirements for studies of this type may differ depending on the setting and we would appreciate some further clarification regarding your research. Could you please provide further details on why your study is exempt from the need for approval and confirmation from your institutional review board or research ethics committee (e.g., in the form of a letter or email correspondence) that ethics review was not necessary for this study? Please include a copy of the correspondence as an ""Other"" file.

"I have read the journal's policy and the authors of this manuscript have the following competing interests: Kenji Kawashima is employed by RIVERFIELD and owns stock in the company. Takahiro Kannno is employed by RIVERFIELD and serves as CTO. Harue Akasaka and other co-authors have no conflict of interest."

Reviewers' comments:

Reviewer's Responses to Questions

**Comments to the Author**

1. Is the manuscript technically sound, and do the data support the conclusions?

Reviewer #1: Yes

Reviewer #2: Partly

2. Has the statistical analysis been performed appropriately and rigorously? 

Reviewer #1: Yes

Reviewer #2: I Don't Know

3. Have the authors made all data underlying the findings in their manuscript fully available?

Reviewer #1: Yes

Reviewer #2: Yes

4. Is the manuscript presented in an intelligible fashion and written in standard English?

Reviewer #1: Yes

Reviewer #2: Yes

5. Review Comments to the Author

Reviewer #1: This manuscript explores the impact of narrow communication bandwidth on surgical operation and surgeon fatigue during remote surgery. I appreciate the fact that you are simulating a real remote surgery, and the manuscript is generally well-written.

The following are some questions to ask.

1) Is there a clear reason why you chose surgical residents instead of skilled surgeons as subjects?

2) It is stated that there was no difference in communication delay. In this study, did you observe any overshoot of forceps manipulation due to communication delay?

3) Even though the task completion time was extended, the subjects judged that the image degradation by packet loss did not affect the surgery, and this part is very confusing. Isn't the extended task completion time caused by a decrease in image quality, if there is no communication delay? You should be able to describe the issues you have considered more clearly.

4) This study has an important implication on the impact of narrow-banding due to network congestion and interference on tele-surgery. Based on the premise that ultra-high speed, high capacity, low latency, and multiple connections in communications will rapidly advance, could you suggest another implication of this paper for the future of tele-surgery?

Reviewer #2: The authors reports combined communication network environment on telerobotic surgery performance and surgeon fatigue.

I recommend to address following points for the paper more informative.

Overall, it is difficult to evaluate the results obtained by surgical residents without prior robotic surgery experience.

It seems that factors other than the communication environment had a significant impact on the results.

It seems that same results are obtained when the same procedure is performed by a skilled surgeon？

Authors should describe the information whether packet loss is more likely to occur in the case of motion and images (e.g., far field or near field).

There is any difference in the accuracy of sutures performed at 1 Gbps or 3 Mbps?

Please provide a representative image of each of different communication speeds.

6. PLOS authors have the option to publish the peer review history of their article (what does this mean?). If published, this will include your full peer review and any attached files.

Reviewer #1: No

Reviewer #2: No

---

## [Author Response · Author response to Decision Letter 0]

8 Apr 2022

Responses to Reviewers

Responses to the Comments by the Associate editor:

Reply:

Thank you for your help. We have consulted the template and we are confident that our revised manuscript meets your style requirements.

2. You indicated that ethical approval was not necessary for your study. We understand that the framework for ethical oversight requirements for studies of this type may differ depending on the setting and we would appreciate some further clarification regarding your research. Could you please provide further details on why your study is exempt from the need for approval and confirmation from your institutional review board or research ethics committee (e.g., in the form of a letter or email correspondence) that ethics review was not necessary for this study? Please include a copy of the correspondence as an ""Other"" file.

Reply:

Thank you very much for this important comment. We have attached a confirmation letter from our ethical committee as an “Other” file.

3. Thank you for stating the following in the Competing Interests section: "I have read the journal's policy and the authors of this manuscript have the following competing interests: Kenji Kawashima is employed by RIVERFIELD and owns stock in the company. Takahiro Kannno is employed by RIVERFIELD and serves as CTO. Harue Akasaka and other co-authors have no conflict of interest."

Please confirm that this does not alter your adherence to all PLOS ONE policies on sharing data and materials, by including the following statement: "This does not alter our adherence to PLOS ONE policies on sharing data and materials.” (as detailed online in our guide for authors http://journals.plos.org/plosone/s/competing-interests ). If there are restrictions on sharing of data and/or materials, please state these. Please note that we cannot proceed with consideration of your article until this information has been declared.

Reply:

Thank you for your help. We have included the following statement: "This does not alter our adherence to PLOS ONE policies on sharing data and materials.” As requested, our Competing Interest statement also appears in our new cover letter. Thank you for changing the online submission form for us.

Reply:

Thank you for your kind suggestion. The corresponding author has an ORCID iD and configured it to be enabled in Editorial manager.

ORCID iD, https://orcid.org/0000-0001-6513-1202

Responses to the Comments by Reviewer 1:

1) Is there a clear reason why you chose surgical residents instead of skilled surgeons as subjects?

Reply:

Thank you very much for your important question. Telesurgery is destined to be further developed in clinical settings in the near future; many surgeons, including young surgeons, as well as experts, will be involved. We work at a teaching hospital, so we thought that young surgeons, who are the core players in the surgical field of the future, should evaluate this new surgical system and clarify the issues they deem meaningful. This is the reason why we chose residents as subjects for this study.

2) It is stated that there was no difference in communication delay. In this study, did you observe any overshoot of forceps manipulation due to communication delay?

Reply:

Thank you for your question. Overshooting due to communication delay was not observed. We evaluated it using GEARS which has an evaluation item that includes overshoot of forceps called “Depth perception.” There was no significant difference in “Depth perception” by line (3.7 vs. 3.5, p = 0.35). We have added text to the Results section to clarify this. (Page 16-17, Lines 267-269). 

3) Even though the task completion time was extended, the subjects judged that the image degradation by packet loss did not affect the surgery, and this part is very confusing. Isn't the extended task completion time caused by a decrease in image quality, if there is no communication delay? You should be able to describe the issues you have considered more clearly.

Reply:

Thank you very much for your invaluable comments. The prolonged task completion time was certainly due to the degraded image quality. However, according to their subjective evaluation, the subjects evaluated it as not affecting the task. This suggests that there is a certain acceptable range of environment in which the surgeons can perform the tasks. In other words, the surgeon adapts to the degradation of the image and unconsciously operates slightly slower to pursue accuracy and complete the task, so the surgeon feels that surgical operation, itself, is unaffected. Objectively speaking, yes, the task completion time is extended. Therefore, a discrepancy between objective and subjective assessments exists. We have added text to the Discussion section (Page 24-25, Lines 389-396) to address this phenomenon.

4) This study has an important implication on the impact of narrow-banding due to network congestion and interference on tele-surgery. Based on the premise that ultra-high speed, high capacity, low latency, and multiple connections in communications will rapidly advance, could you suggest another implication of this paper for the future of tele-surgery?

Reply:

Thank you very much for your helpful recommendations. We have added the following implication to the Discussion section (Page 26, Lines 414-422).

: “In the near future, the communication environment is likely to advance rapidly, telesurgery situations will evolve and become more stabilized with fewer communication interruptions. However, there is no doubt that determining the minimum environment necessary for safe and economic telesurgery is necessary. Furthermore, the state of telecommunications is not always adequate in the very regions of the world where telesurgery is most needed. In order to achieve both objective and subjective evaluations of telesurgery protocols, it is necessary to set appropriate standards for acceptable packet loss and video quality, because some discrepancy between subjective and objective evaluations is always likely to exist.”

Responses to the Comments by the reviewer 2: 

Overall, it is difficult to evaluate the results obtained by surgical residents without prior robotic surgery experience. It seems that factors other than the communication environment had a significant impact on the results. It seems that same results are obtained when the same procedure is performed by a skilled surgeon?

Reply:

Thank you very much for your invaluable comments. In fact, the residents had trained and practiced robotic surgery prior to this study and had experienced actual robotic surgery as assistants. We believe that they were capable of evaluating the telesurgery environment in the study in contrast to actual robotic surgery. However, as Reviewer 2 pointed out, such factor may be viewed as limitations of this study. The acceptable range of image quality degradation and communication delay could be quite different for experts. We have added a description of the subject's experience of robotic surgery to the Methods section (Page 9, 145).

Authors should describe the information whether packet loss is more likely to occur in the case of motion and images (e.g., far field or near field).

Reply:

Thank you very much for your excellent suggestion. In this study, more packet loss occurred when robot operation was performed. This is because the communication environment of this study was set to reserve 1Mbps for robot operations, and the bandwidth was reduced by 1Mbps when robot operation was started. However, we are afraid to say that we cannot comment on exactly where it affects the field of view, the far or near side, because we evaluated image changes in terms of whole images.

There is any difference in the accuracy of sutures performed at 1 Gbps or 3 Mbps?

Reply:

Thank you for your question. The accuracy of sutures was equal at each bandwidth. We evaluated it using GEARS, which has an evaluation item that includes the accuracy of the movement of the instruments called “Depth perception.” There was no significant difference “Depth perception” by line (3.7 vs. 3.5, p = 0.35). We have added text to the Results section. (Page 16-17, Lines 267-269). 

Please provide a representative image of each of different communication speeds.

Reply:

Thank you very much for your important comments. We regret to say that we did not record individual images for the various stages of the procedures. Instead, we evaluated image quality using an image quality evaluation system as shown in Figure 5 to indicate how image quality changed and whether or not it affected the procedures.

---

## [Decision Letter · Decision Letter 1]

28 Apr 2022

PONE-D-22-03175R1Impact of the suboptimal communication network environment on telerobotic surgery performance and surgeon fatiguePLOS ONE

Dear Dr. Hakamada,

Thank you for submitting your manuscript to PLOS ONE. After careful consideration, we feel that it has merit but does not fully meet PLOS ONE’s publication criteria as it currently stands. Therefore, we invite you to submit a revised version of the manuscript that addresses the points raised during the review process.

We look forward to receiving your revised manuscript.

Kind regards,

Norikatsu Miyoshi, M.D., Ph.D., FACS

Academic Editor

PLOS ONE

Journal Requirements:

Reviewers' comments:

Reviewer's Responses to Questions

**Comments to the Author**

1. If the authors have adequately addressed your comments raised in a previous round of review and you feel that this manuscript is now acceptable for publication, you may indicate that here to bypass the “Comments to the Author” section, enter your conflict of interest statement in the “Confidential to Editor” section, and submit your "Accept" recommendation.

Reviewer #1: All comments have been addressed

Reviewer #2: All comments have been addressed

2. Is the manuscript technically sound, and do the data support the conclusions?

Reviewer #1: Yes

Reviewer #2: Partly

3. Has the statistical analysis been performed appropriately and rigorously? 

Reviewer #1: Yes

Reviewer #2: I Don't Know

4. Have the authors made all data underlying the findings in their manuscript fully available?

Reviewer #1: Yes

Reviewer #2: Yes

5. Is the manuscript presented in an intelligible fashion and written in standard English?

Reviewer #1: Yes

Reviewer #2: Yes

6. Review Comments to the Author

Reviewer #1: Thank you for your response to my comments.

In response to the second comment, "We have added text to the Results section to clarify this.(Page 16-17, Lines 267-269)", but unfortunately I could not find the changed section.

Reviewer #2: (No Response)

7. PLOS authors have the option to publish the peer review history of their article (what does this mean?). If published, this will include your full peer review and any attached files.

Reviewer #1: No

Reviewer #2: No

---

## [Author Response · Author response to Decision Letter 1]

10 May 2022

Responses to the Comments by the Associate Editor:

Reply:

Thank you very much for your helpful comment. We confirmed that the reference list is complete and correct. Also, there was no retracted reference.

Responses to the Comments by Reviewer 1:

1) Thank you for your response to my comments.

In response to the second comment, "We have added text to the Results section to clarify this.(Page 16-17, Lines 267-269)", but unfortunately I could not find the changed section.

Reply:

Thank you for pointing this out. We are very sorry that there was an incorrect statement in the description of the modification. We revised the description regarding overshooting as follows.

Overshooting due to communication delay was not observed. We evaluated it using GEARS, which has an evaluation item that includes the concept of overshooting of forceps called “Depth perception.” There was no significant difference in “Depth perception” by line (3.7 vs. 3.5, p = 0.35). Furthermore, we have added text to the Results section to clarify this (Page 17, Lines 270-272).

---

## [Decision Letter · Decision Letter 2]

3 Jun 2022

Impact of the suboptimal communication network environment on telerobotic surgery performance and surgeon fatigue

PONE-D-22-03175R2

Dear Dr. Kenichi Hakamada,

We’re pleased to inform you that your manuscript has been judged scientifically suitable for publication and will be formally accepted for publication once it meets all outstanding technical requirements.

Kind regards,

Norikatsu Miyoshi, M.D., Ph.D., FACS

Academic Editor

PLOS ONE

Additional Editor Comments (optional):

Reviewers' comments:

Reviewer's Responses to Questions

**Comments to the Author**

1. If the authors have adequately addressed your comments raised in a previous round of review and you feel that this manuscript is now acceptable for publication, you may indicate that here to bypass the “Comments to the Author” section, enter your conflict of interest statement in the “Confidential to Editor” section, and submit your "Accept" recommendation.

Reviewer #1: All comments have been addressed

2. Is the manuscript technically sound, and do the data support the conclusions?

Reviewer #1: Yes

3. Has the statistical analysis been performed appropriately and rigorously? 

Reviewer #1: Yes

4. Have the authors made all data underlying the findings in their manuscript fully available?

Reviewer #1: Yes

5. Is the manuscript presented in an intelligible fashion and written in standard English?

Reviewer #1: Yes

6. Review Comments to the Author

Reviewer #1: (No Response)

7. PLOS authors have the option to publish the peer review history of their article (what does this mean?). If published, this will include your full peer review and any attached files.

Reviewer #1: No

---

## [Editor Report · Acceptance letter]

9 Jun 2022

PONE-D-22-03175R2 

Impact of the suboptimal communication network environment on telerobotic surgery performance and surgeon fatigue 

Dear Dr. Hakamada:

I'm pleased to inform you that your manuscript has been deemed suitable for publication in PLOS ONE. Congratulations! Your manuscript is now with our production department. 

Kind regards, 

on behalf of

Dr. Norikatsu Miyoshi 

Academic Editor

PLOS ONE